# Single-cell RNA sequencing of nc886, a non-coding RNA transcribed by RNA polymerase III, with a primer spike-in strategy

**Gyeong-Jin Shin**[1,2☯], **Byung-Han Choi**[3☯], **Hye Hyeon Eum**[1], **Areum Jo**[1], **Nayoung Kim**[1], **Huiram Kang**[1,2], **Dongwan Hong**[2,4], **Jiyoung Joan Jang**[3], **Hwi-Ho Lee**[3], **Yeon-Su Lee**[5], **Yong Sun Lee**[3]*, **Hae-Ock Lee**[1,2]*

1 Department of Microbiology, The Catholic University of Korea, Seoul, Korea, 2 Department of Biomedicine and Health Sciences, The Catholic University of Korea, Seoul, Korea, 3 Department of Cancer Biomedical Science, Graduate School of Cancer Science and Policy, National Cancer Center, Goyang, Korea, 4 Department of Medical Informatics, The Catholic University of Korea, Seoul, Korea, 5 Division of Rare Cancer, Research Institute, National Cancer Center, Goyang, Korea

☯ These authors contributed equally to this work.
* yslee@ncc.re.kr (YSL); haeocklee@catholic.ac.kr (HOL)

**Data Availability Statement:** The sequencing data is publicly accessible on the National Center for Biotechnology Information (NCBI) Sequence Read

## Abstract

Single-cell RNA sequencing (scRNA-seq) has emerged as a versatile tool in biology, enabling comprehensive genomic-level characterization of individual cells. Currently, most scRNA-seq methods generate barcoded cDNAs by capturing the polyA tails of mRNAs, which exclude many non-coding RNAs (ncRNAs), especially those transcribed by RNA polymerase III (Pol III). Although previously thought to be expressed constitutively, Pol III-transcribed ncRNAs are expressed variably in healthy and disease states and play important roles therein, necessitating their profiling at the single-cell level. In this study, we developed a measurement protocol for nc886 as a model case and initial step for scRNA-seq for Pol III-transcribed ncRNAs. Specifically, we spiked in an oligo-tagged nc886-specific primer during the polyA tail capture process for the 5'scRNA-seq. We then produced sequencing libraries for standard 5' gene expression and oligo-tagged nc886 separately, to accommodate different cDNA sizes and ensure undisturbed transcriptome analysis. We applied this protocol in three cell lines that express high, low, and zero levels of nc886. Our results show that the identification of oligo tags exhibited limited target specificity, and sequencing reads of nc886 enabled the correction of non-specific priming. These findings suggest that gene-specific primers (GSPs) can be employed to capture RNAs lacking a polyA tail, with subsequent sequence verification ensuring accurate gene expression counting. Moreover, we embarked on an analysis of differentially expressed genes in cell line sub-clusters with differential nc886 expression, demonstrating variations in gene expression phenotypes. Collectively, the primer spike-in strategy allows combined analysis of ncRNAs and gene expression phenotype.

Archive (SRA) under the BioProejct ID PRJNA1124055. The corresponding data in the Gene Expression Omnibus (GEO) is available under the accession number GSE269878.

**Funding:** This study was supported by the National Research Foundation of Korea (NRF) funded by the Ministry of Science, ICT & Future Planning (RS-2023-00220840 awarded to to HL). Grants from the National Cancer Center, Korea (NCC-2210320 and NCC-2311362 awarded to YSL and NCC-2210360 to Y-SL). Basic Medical Science Facilitation Program, through the Catholic Medical Center of the Catholic University of Korea funded by the Catholic Education Foundation and KREONET/GLORIAD service provided by KISTI (Korea Institute of Science and Technology Information) awarded to HL. The funders had no role in study design, data collection and analysis, decision to publish, or preparation of the manuscript.

**Competing interests:** The authors have declared that no competing interests exist.

## Introduction

Non-coding RNAs (ncRNAs) and next-generation sequencing (NGS) technologies have been among the greatest advances in biology over the past two decades [1, 2]. Numerous studies have documented the diverse biological roles of ncRNAs, with the most prominent being the gene-regulatory functions of microRNAs and long ncRNAs (lncRNAs). NGS techniques, which offer unprecedented high-throughput capabilities, have generated enormous amounts of genomic, epigenomic, and transcriptomic data. Additionally, NGS has greatly advanced the field of ncRNAs by enabling the capture of low-copy RNAs, many of which have been identified to be non-coding [3]. More recently, NGS has been applied at the single-cell level.

Single-cell RNA sequencing (scRNA-seq) technologies continue to advance, with the fundamental principle being the creation of barcoded cDNAs that allow differentiation of individual cells [4, 5]. Particularly, droplet-based approaches such as the Chromium system (10X genomics) provide a combination of simplicity and cost-effectiveness, and they currently constitute the majority of scRNA-seq data [6]. A limitation of the system arises from the use of oligo-dT sequences during cDNA synthesis, restricting the focus to mRNAs with a polyA tail. Analogous constraints have also been observed in bulk RNA sequencing analysis. Consequently, various strategies are employed to investigate RNA molecules that are typically excluded such as non-polyAed lncRNAs and small RNAs [7]. In bulk RNA sequencing, adaptor ligation or random primers may be applied following the removal of ribosomal RNAs (rRNAs) [8, 9]. The addition of polyA or polyU tails also allows sequencing of non-polyAed RNAs [10]. Several research groups have reported methods for quantifying non-polyAed RNAs at the single-cell level [11–13]. These methods modify techniques used in total RNA-seq in bulk, including random priming, polyA tailing, and/or rRNA removal. While showing promise, they are developed for C1 microfluidic devices or SMART-seq, which are costly compared with droplet-based methods.

Despite the increase in research on ncRNAs, medium-sized ncRNAs—a subset of ncRNAs transcribed by RNA polymerase III (Pol III)—remain underexplored. These RNAs include transfer RNAs (tRNAs), 5S rRNA, and U6 small nuclear RNA, and their roles are so fundamental that it is challenging to imagine their dynamic expression. Therefore, Pol III-transcribed ncRNAs (Pol III-ncRNAs) have attracted minimal attention during the application and analysis of NGS. However, this view of Pol III-ncRNAs has recently changed. The repertoire of Pol III-ncRNAs is more diverse than previously thought [14], and Pol III transcriptomes vary depending on the biological situation [15]. The best examples of dynamically expressed Pol III-ncRNAs are nc886 and tRNA-derived RNA fragments, which control gene expression [16, 17]. Thus, it is essential to obtain Pol III transcriptomes and analyze them in comparison to other -omics data.

As an initial attempt to classify Pol III transcriptomes, we developed a protocol for measuring nc886 in droplet-based scRNA-seq. We chose nc886 because [18] 1) it is transcribed from a single genomic locus, unlike most other Pol III genes, which have identical or highly similar sequences scattered at multiple loci across the genome; 2) it undergoes no post-transcriptional modifications, unlike tRNAs; and 3) its expression is abundant in some cancer cells but is absent in others. These features provide unambiguity in mapping and a set of cell lines for comparison, making nc886 an ideal ncRNA for establishing a new sequencing protocol. Furthermore, given its important roles in cancer and immunity, a single-cell expression profile of nc886 will provide valuable information.

## Materials and methods

### Cell culture, RNA isolation, and qRT-PCR

WPMY-1 and Hep3B cell lines were purchased from the American Type Culture Collection (Manassas, VA). The HEK293T line was from our laboratory stock. We produced

nc886-expressing Hep3B cells (designated "Hep3B-886" hereafter) using a lentiviral plasmid, "pLL3.7.Puro.U6:nc886." This plasmid was derived from the original lentiviral plasmid, pLL3.7 (Addgene, Watertown, MA) and contains the nc886 gene (a 102 nucleotide (nt)-long DNA fragment) under the U6 promoter [19]. Lentivirus production, infection, and selection of puromycin-resistant cells were performed per standard laboratory procedures. From the three cell lines, total RNA was isolated using TRIzol™ Reagent (Invitrogen, Carlsbad, CA), and nc886 was measured by qRT-PCR and northern hybridization as described previously [20].

### nc886-specific primer design and sequencing library construction

For generation of the nc886 feature library, a gene-specific primer (GSP) was designed to flank 3' nc886 sequences with a feature barcode and a sequencing adaptor (Table 1). The sequence used was 5'-<u>CGGAGATGTGTATAAGAGACAG</u>NNNNNNNNNNN*GTATGTCCGCTCGAT*NNNNNNN NNN**AGGGTCAGTAAGCACCCGCG**-3'. The first underlined 22 nucleotides (nts) represent a Read2N adaptor (10X genomics); the second 15 nts italicized, a feature barcode (Total-SeqTM-C0182, BioLegend); and the third 20 nts in bold, complementary 3' nc886 sequences. The three nucleotide blocks are separated by 10- or 9-nts spacer sequences. Reverse transcription using this nc886-GSP generates a 1st-strand cDNA consisting of CCC-nc886-spacer-feature barcode-spacer-Read2N sequences. Second-strand synthesis is accomplished using template switching oligos (TSOs) attached to gel beads containing the 5' scRNA-seq reagent.

### Modified 5' single-cell RNA sequencing and read processing

Single-cell suspensions of WPMY-1, Hep3B-886, and HEK293T cell lines were mixed in equal numbers and subjected to scRNA-seq using Chromium Next GEM Single-Cell V(D)J Reagent Kits v2. We set the cell recovery rate to 5,000 per library and followed the manufacturer's instructions with a slight modification. During the GEM generation and barcoding step, we added 0.1 μM nc886-GSP to the master mix. In addition to the 5' gene expression (GEX) library, the nc886 feature library was constructed using the 5' Feature Barcode Kit (10X Genomics). Both libraries were sequenced on an Illumina Hiseq X as 100-bp paired-ends. Sequencing reads were processed using the Multi-pipeline in the Cell Ranger toolkit (v5.0.0) and mapped to the GRCh38 human reference genome.

### Processing of oligo-tagged sequences

In raw paired-end reads, preprocessing was performed separately for Read 1 (R1) and Read 2 (R2). For R1, reads containing the nc886 sequence were selected using seqkit (command: seqkit grep -s -p <nc886 sequence>) [21], and R2 was filtered for reads containing both the feature barcode sequence and nc886 sequence (command: seqkit grep -s -p <feature barcode sequence> | seqkit grep -s -p <nc886 sequence>). When selecting reads containing the nc886 sequence, the number of allowable mismatches was specified using the -m option (e.g., allowing one mismatch: -m 1). After preprocessing R1 and R2 separately, cases were possible where the read pairs in R1 and R2 did not match. To address this, a custom python script was used on the preprocessed R1 and R2 to extract only the paired reads. In brief, overlapping read IDs between R1 and R2 were extracted, and the corresponding sequence and quality score information for each ID were extracted to generate new paired reads. These new paired reads were subsequently used in downstream analyses.

### DNA isolation and SNP genotyping array

Genomic DNA was extracted from the three cell lines using the PureLinkTM Genomic DNA Mini kit (Invitrogen). Total of 778,783 single nucleotide polymorphisms (SNPs) were

**Table 1. nc886 feature library construction.**

| Step | Name | Sequences | note |
|---|---|---|---|
| reference RNA | nc886 | 5'-CGGGUCGGAGUUAGCUCAAGCGGUUACCUCCUCAUGCCGGACUUUCUAUCUGUCCUAUCUGUGCUGGGGUUCGAGACCCGCGGGUGCUUACUGACCCUUU-3' | |
| primer design | nc886-GSP | 5'-CGGAGATGTGTATAAGAGACAGAGNNNNNNNNNGTATGTCCGCTCGANTNNNNNNNNNNAGGGTCAGTAAGCACCGCG-3' | Read2N(N22), *feature barcode(N15)*, **nc886-specific sequence (N20)** |
| reverse Transcription | nc886 cDNA | 3'-CCCGCCCAGCCTCAATCGAGTTCGCCAATGGAGTTCGCCAATGGAGGAGTACGGCCTGAAAGATAGACAGGTAGAGACACGACCCCAAGCTCTGGGCGCCCACGAATGACTGGGA*NNNNNNNNNTAGCTCGCCTGTATG*NNNNNNNNNNGACAGAGAATATGTGTAGAGGC-5' | |
| cDNA Amplification | amplified cDNA | 5'-CTACACGACGCTCTTCCGATCT-N16-N10-TTTCTTATATGGGCGGGTCGGAGTTAGCTCAAGCGGTTACCTCCTCCATCTCTGTGCTGGGGTTCGAGACCCGCGGGTGCTTACTGACCCT*NNNNNNNNNATCGAGCCGGACATAC*NNNNNNNNNCTGTCTCTTATACACATCTCCG-3'<br><br>3'-GATGTGCTGCGAGAGGCTAGA-N16-N10-AAAGAATATACCCGCCCAGCCTCAATCGAGTTCGCCAATGGAGGAGTACGGCCTGAAAGATAGACAGGTAGAGACACGACCCCAAGCTCTGGGCGCCCACGAATGACTGGGA*NNNNNNNNNTAGCTCGCCTGTATG*NNNNNNNNNNGACAGAGAATATGTGTAGAGGC-5' | |
| library construction | library | 5'-ACACTCTTTCCCTACACGACGCTCTTCCGATCT-N16-N10-TTTCTTATATGGGCGGGTCGGAGTTAGCTCAAGCGGTTACCTCCTCCATCTCTGTGCTGGGGTTCGAGACCCGCGGGTGCTTACTGACCCT*NNNNNNNNNATCGAGCCGGACATAC*NNNNNNNNNCTGTCTCTTATACACATCTCCGAGCCCACGAGAC-3'<br><br>3'-TGTGAGAAAGGGATGTGCTGCGAGAGGCTAGA-N16-N10-AAAGAATATACCCGCCCAGCCTCAATCGAGTTCGCCAATGGAGGAGTACGGCCTGAAAGATAGACACGACCCCAAGCTCTGGGCGCCCACGAATGACTGGGA*NNNNNNNNNTAGCTCGCCTGTATG*NNNNNNNNNNGACAGAGAATATGTGTAGAGGCTCGGGTGCTCTG-5' | cell barcode(N16), UMI (N10) TSO (TTTCTTATGGG) |

genotyped on the Infinium Global Screening Array MG v3.0 (Illumina, San Diego, CA) by Macrogen (Seoul, Korea) following standard Illumina procedures. Normalized signal intensity and genotype data were computed using the Illumina/BeadArray Files Python library. The variant calling format (VCF) genotype file was generated using the GRCh38 reference genome.

## Demultiplexing

To demultiplex the data from three cell lines pooled in scRNA-seq, we followed the freemuxlet workflow (http://github.com/statgen/popscle) [22]. Briefly, the popscle tool dsc-pileup was run on the BAM file generated by the Cell Ranger toolkit alongside a reference VCF file downloaded from Demuxafy (https://demultiplexing-doublet-detectingdocs.readthedocs.io/en/latest/index.html). This tool enhances accuracy in multiple demultiplexing and doublet detection methods. Subsequently, the freemuxlet tool, set to its default parameters, was used to deconvolve the sample identities. Each of the three cell lines (HEK293T, Hep3B-886, WPMY-1) had a distinct VCF file containing chromosomal position information. The cell lines were distinguished based on the similarities between freemuxlet-annotated genotypes and genotypes detected by SNP arrays. During this step, doublets (DBL) and ambiguous (AMB) barcodes were removed (Excluded AMB+DBL: 4,172 cells; HEK293T: 1,687 cells; Hep3B-886: 2,001 cells; WPMY-1: 4,738 cells).

## Single-cell RNA sequencing analysis using Seurat

From the Cell Ranger outputs, the raw gene-cell-barcode matrix was processed using the Seurat v4.2.2 R package [23]. Low-quality cells were filtered with the criteria nCount>2000 and percent. mito <15. Potential multiplets were predicted by Scrublet and removed [24]. After QC filtering, the unique molecular identifier (UMI) count matrix was log-normalized and scaled using a z-transform. Utilizing the PC ElbowPlot function of Seurat, PC 7 was selected as a distinct subset of principal components. Cell clustering and Uniform Manifold Approximation and Projection (UMAP) visualization were then conducted using the 'FindClusters' and 'RunUMAP' functions. The resolution was set to 0.3 or 0.6, segregating three or six clusters, respectively.

## Pathway enrichment analysis and data visualization

Subcluster analysis for WPMY-1 cells was conducted using the 'enrichGO' function of the 'clusterProfiler' R package (version 4.6.2), focusing on the top differentially expressed genes (DEGs). Genes were filtered based on an adjusted p-value and q-value (< 0.05). The 'org.Hs. eg.db' annotation package (version 3.15.0) was used for organism-specific categorization. Data filtering was applied to GeneRatios greater than 0.10, and results were organized in ascending order of adjusted p-values, specifically targeting the 'biological process' category in the Gene Ontology.

## Results

### Generation of the nc886 feature library using an oligo-tagged gene-specific primer

To assess the feasibility of using a GSP during droplet-based scRNA-seq procedures (10x genomics chromium system), we selected nc886 as the model gene and chose three cell lines with varying levels of nc886 expression: WPMY-1, Hep3B-886, and HEK293T. Hep3B-886 is an nc886-expressing stable cell line derived from a hepatocellular carcinoma cell line, Hep3B [25]. The difference in nc886 expression levels among these cell lines was clearly demonstrated by two methods: qRT-PCR and northern hybridization (Fig 1A).

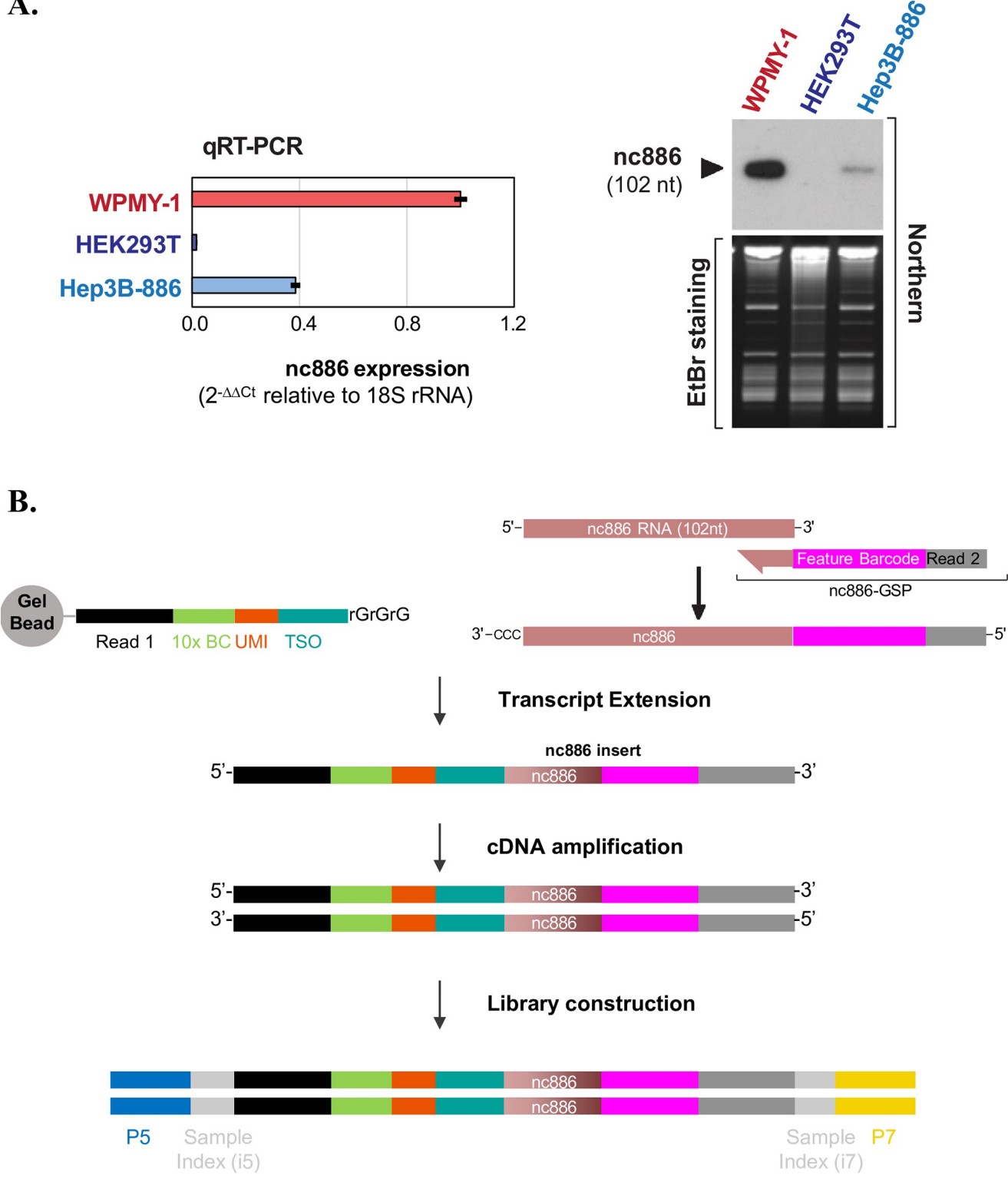

**Fig 1. Modification of 5' scRNA-seq for nc886 detection.** (A) nc886 gene expression levels in three cell lines, measured by qRT-PCR (left panel) and by Northern hybridization (right panel). In qRT-PCR, each bar represents an average of triplicate samples, with the standard deviation indicated. Primer sequences (5' to 3') are: cgggtcggagttagctcaagcgg (forward primer for nc886), aagggtcagtaagcacccgcg (reverse primer and Northern probe for nc886), cggctttggtgactctagat (forward primer for 18S rRNA) and gcgactaccatcgaaagttg (reverse primer for 18S rRNA). (B) A cartoon depicting the procedure for library preparation in which nc886 gene-specific primer (nc886-GSP) was spiked-in. Diagrams are drawn to show nc886-GSP and products (whose actual sequences are listed in Table 1) in each step. The final library contains sample index and sequencing adaptors P5 and P7.

To capture nc886 transcripts without a polyA tail, we used a GSP with additional feature barcode and adaptor sequences (Fig 1B and S1 Fig). The modifications implemented in the GEM generation and the barcoding reaction mix are detailed in the Methods section. Addition of the GSP allows extension of the nc886 transcript ("nc886-GSP" in Fig 1B and S1A Fig), yielding cDNA containing the nc886 sequence flanked by adaptors to enable library construction. According to our design, the resulting nc886 feature barcode library was expected to contain Read 1 sequences, 10x cell barcode, UMI, and TSO at the 5' end as well as 15 nts-feature barcode and 'Read 2' sequences (Fig 1B and S1A Fig). Before applying to scRNA-seq, we validated this approach *in vitro* by synthetizing cDNA with nc886-GSP and the tagged TSO from total RNA isolated from an nc886-expressing cell line (S1A Fig, left side). PCR amplification of the cDNA yielded a single product at the expected size (S1B and S1C Fig). The PCR product was further confirmed by sequencing (data not shown). Together, these data indicated that the nc886 cDNA was correctly made as we designed.

## Determining cell line identities using SNP and gene expression profiles

In the modified scRNA-seq experiment, we pooled the three cell lines with differential nc886 expression to generate multiplex data. In parallel, a 5' scRNA-seq library for gene expression analysis was produced as a separate sequencing material. WPMY-1 is a myofibroblast cell line derived from a prostate cancer patient [26]. Hep3B, the original cell line of Hep3B-886, was derived from liver cancer with epithelial morphology and hepatitis B virus integration. The HEK293T cell line originated from the human embryonic kidney [27, 28].

The first step in our data analysis was to systematically assign pooled scRNA-seq data into respective cell lines using SNP patterns. This segregation was a prerequisite for the investigation of phenotypic alterations in gene expression, especially regarding nc886 expression levels. We used Freemuxlet, a tool recommended by the 10x Genomics Analysis Guide (https://www.10xgenomics.com/resources/analysis-guides/bioinformatics-tools-for-sample-demultiplexing), to categorize cells into three distinct clusters. Overlap between SNPs detected in genomic DNA and scRNA-seq data provided cell line identity for each SNP cluster (Fig 2A). Cells corresponding to doublets and ambiguous categories in SNP expression were excluded from subsequent analyses.

In the second step, we performed clustering analysis based on the 5' gene expression after further quality control (QC) filtration to select cells with a minimum UMI count of 2,000, a minimum gene count of 200, and a maximum mitochondrial gene proportion of 15% (Fig 2B, left). Adhering to these criteria, we obtained three clusters, assigned as WPMY-1 cells (1,886 cells), Hep3B-886 (1,782 cells), and HEK293T (1,457 cells) (CLUST0, 1, and 2, respectively, in the right panel of Fig 2B). Comparison of DEGs in the clusters revealed gene expression characteristics of the three cell lines (Fig 2C). Cluster 0 showed prominent expression of mesenchymal genes such as *COL1A1*, *SPARC*, and *CCN1*, which characterize WPMY-1 cells of myofibroblast origin [29]. In the Hep3B-886 cluster, liver-specific genes such as *ALB*, *RBP4*, and *AHSG* were highly expressed [30, 31]. In the cluster of nc886-silenced HEK293T cells, high expression levels of *XIST*, *TSC22D3*, and *RPS4X* were noted. These expression patterns were consistent with those observed in the original parental cell line [32]. These gene expression characteristics confirmed the successful implementation of multiplexing and demultiplexing strategies in scRNA-seq analysis.

## Assessing nc886 gene expression using feature barcoding and sequence alignment strategies

Next, we estimated nc886 expression levels by counting the reads aligned to the feature barcode using the CellRanger multi pipeline (Fig 3A, left, green box). When this method was

## A.

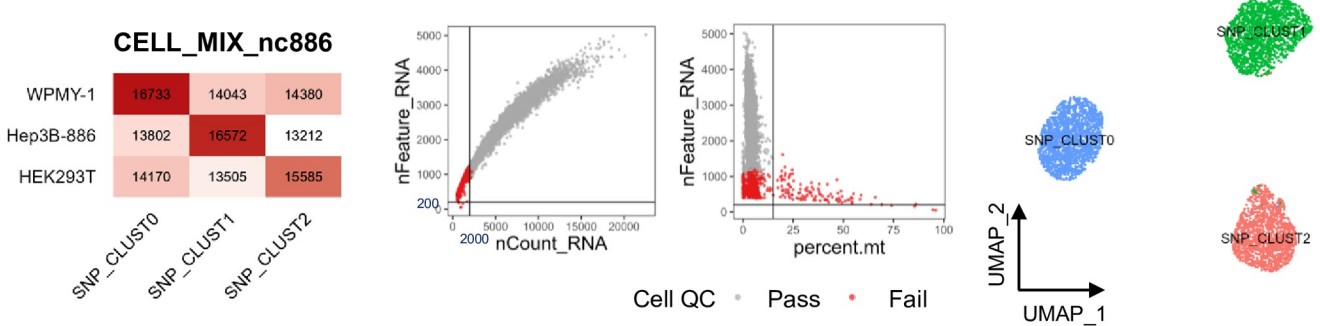

## B.

## C.

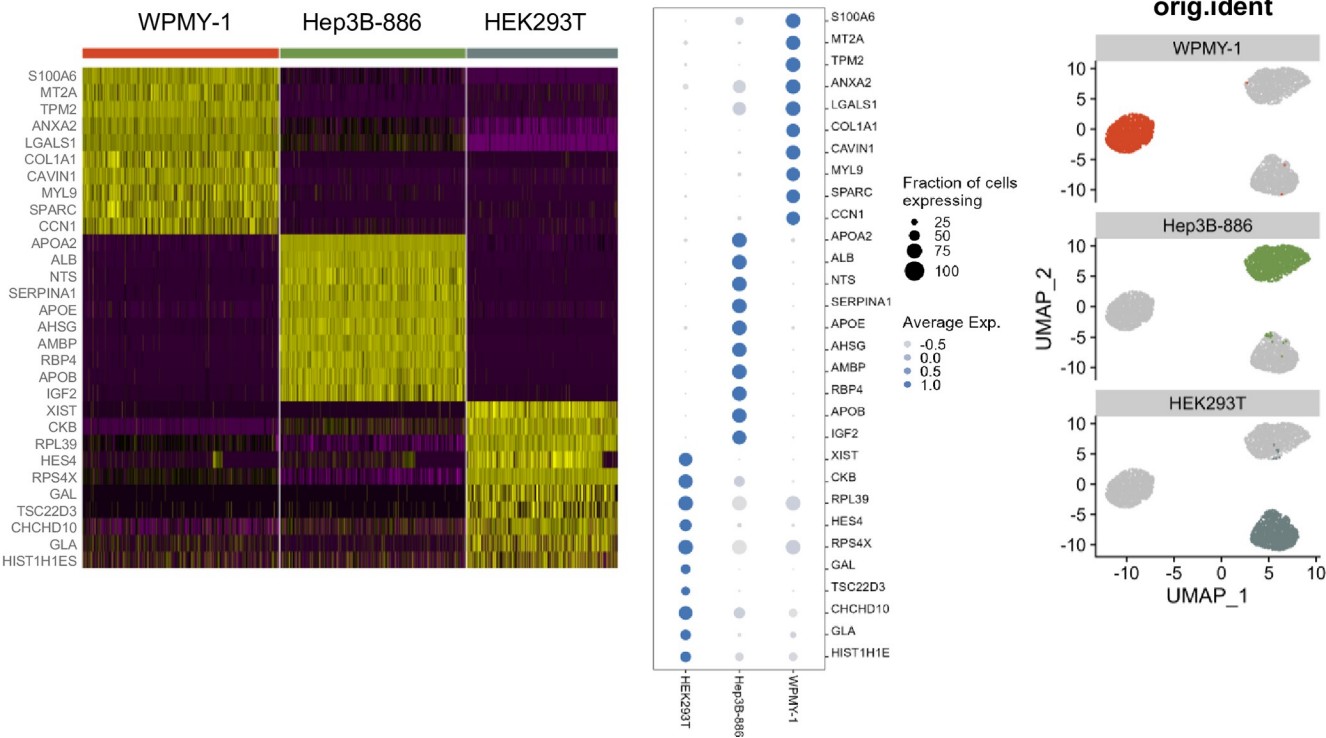

**Fig 2. Gene expression profiling of each cell line.** (A) Cells were classified into three distinct clusters using 'freemuxlet'. Number of overlapping SNPs between the clusters and SNP arrays for each cell line are indicated. Based on the SNP expression, doublets and ambiguous cells are excluded. (B) After freemuxlet runs and doublet/ambiguous cell removal, additional QC filtration was applied: UMI counts(nCount_RNA>2,000), number of genes expressed (nFeature_RNA>200), and proportions of mitochondrial gene expression (percent.mt<15) (left). The UMAP shows SNP clusters within 3 Seurat clusters: SNP_CLUST0 = "WPMY-1", SNP_CLUST1 = "Hep3B-886", SNP_CLUST2 = "HEK293T" (right). (C) DEG analysis showing gene expression characteristics for each cell line, shown as Heatmap (left) and DotPlot (middle). UMAP cluster designation to each cell line (right).

applied for nc886 estimation, we found unexpectedly high nc886 expression in the silenced HEK293T cell cluster (Fig 3B). To investigate this inconsistency, we simulated the nc886 scRNA-seq protocol *in vitro* to identify a potential cause. During our nc886 scRNA-seq, the initially synthesized cDNA was amplified using adaptor sequences outside of nc886 before sequencing. To accurately replicate the nc886 scRNA-seq protocol, we included identical cDNA amplification steps before PCR (S1A Fig, right side). PCR produced fuzzy bands (S1B and S1C Fig), indicating compromised primer specificity.

Upon recognizing this problem, we switched the mapping strategy to process Read 2 from the feature barcode library and to specifically extract nc886-aligned reads (Fig 3A, left, green and blue boxes). Using nc886-specific sequences from the feature barcode data dramatically reduced the total number of reads (Fig 3A, right). These data indicated that non-specific priming, suggested by our *in vitro* experiment (S1 Fig), occurred during the nc886 scRNA-seq. The problem of reduced read numbers was alleviated by allowing a single nucleotide mismatch. This refined procedure—extracting nc886 sequences from Read 2 with up to 1 nt mismatch—yielded results aligned well with the known nc886 expression levels: they were markedly higher in WPMY-1, lower in Hep3B-886, and absent in HEK293T (Fig 3C).

### Integrated analysis demonstrating diversity in gene expression patterns and nc886 levels

After demonstrating the successful detection of nc886 transcripts, we performed an integrated analysis of nc886 and 5' gene expression. First, we re-performed clustering analysis with a higher resolution to find sub-populations with differential gene expression. Clustering in the UMAP space revealed the presence of two distinct clusters for each cell line (Fig 4A, upper UMAPs and a heat map). Subsequently, we assessed whether the cluster separation reflects differential cell cycle phases (Fig 4A, lower UMAP and bar graph). Hep3B-886 (Hep 1 and Hep 2) and HEK293T cell (HEK 1 and HEK 2) clusters showed different cell cycle distribution between clusters. In contrast, WPMY-1 clusters (W1 and W2) manifested similar cell cycle phases, indicating these sub-clusters are not the product of cell cycle phases.

Thereafter, we focused on the WPMY-1 cell line and performed DEG analysis using the Wilcoxon Rank Sum test (Fig 4A, right). Comparison of nc886 expression levels between W1 and W2 clusters showed enrichment of nc886 high cells in the W2 cluster (Fig 4B). In the W2 cluster, DEGs include *POSTN*, *MFAP4*, *DCN*, and *LUM* (Fig 4C), which are closely involved in extracellular matrix organization as well as in cancer invasiveness [33, 34]. By comparison, the W1 cluster DEGs contained *CAV1*, *MT2A*, *KCNMA1*, and *CCND1* curated in response to the metal ion pathway, as well as *FABP5*, *SPHK1*, and *CCN* for regulation of lipid metabolism [35–38]. Consistently, Gene Set Enrichment Analysis (GSEA) annotated protein folding and stability for W1 DEGs and extracellular matrix organization and stimulus for W2 DEGs (Fig 4D). Overall, this combined analysis demonstrated heterogeneity of gene expression phenotypes associated with nc886 levels and suggested that the process may be applicable to other samples with variable nc886 levels, such as colon cancer, to determine the functional impact of nc886 expression.

## Discussion

### Validating a modified scRNA-seq method for non-polyA ncRNA analysis

In this study, we implemented a straightforward strategy to detect nc886 by a gene-specific primer alongside oligo dT during first-strand cDNA synthesis in a 5' scRNA-seq protocol. This approach enables the simultaneous analysis of non-polyA-tailed nc886 and poly(A)

**A.**

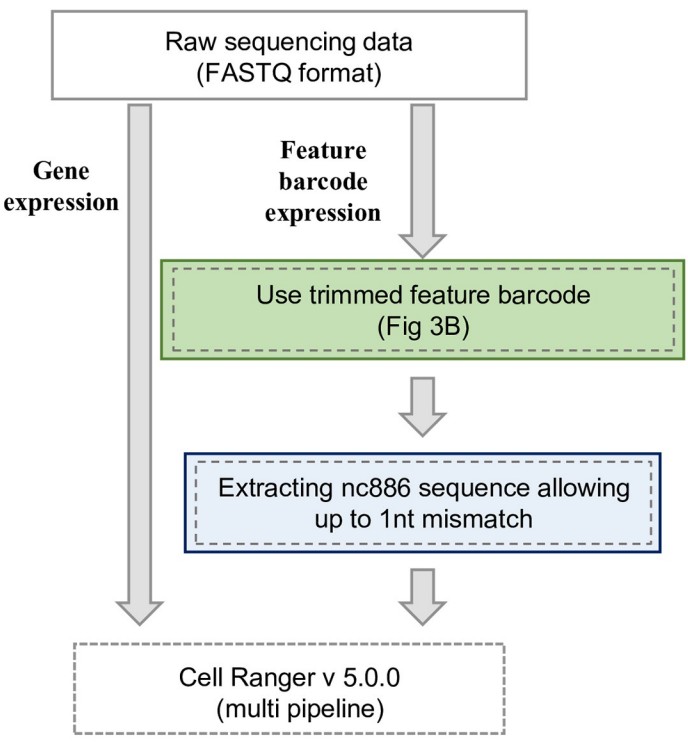

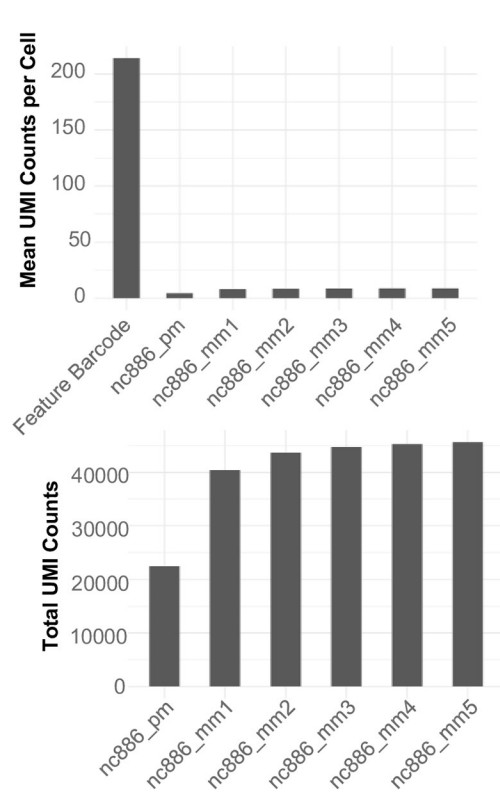

**B.**

**C.**

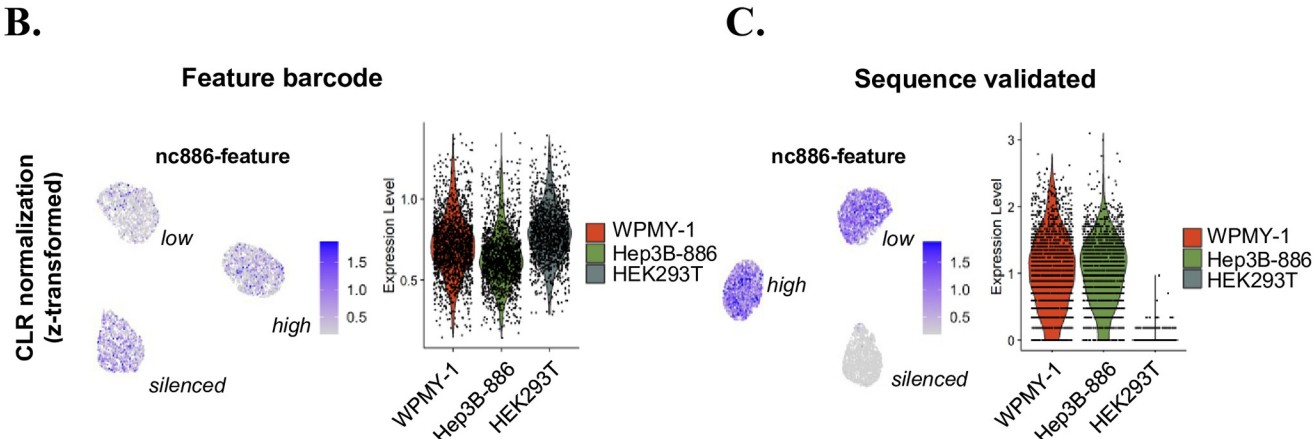

**Fig 3. Assessment of nc886 expression level employing feature barcoding and a sequence alignment strategy.** (A) Overview of the estimation of nc886 gene expression in comparison to transcriptome analysis (left). To assess nc886 gene expression, the feature barcode or nc886 sequence aligned reads were counted. nc886 gene expression and transcriptome data were processed by the CellRanger multi pipeline. Bar plots quantify UMI counts for the feature barcode or nc886 sequence alignments, with mismatch counts ranging from one to five nucleotides: "nc886_pm" and "nc886_mm1-5" denote perfect match to nc886 and the number of mismatches (mm) respectively (right). (B) The expression level of nc886 across cell lines in z-transform values of the feature barcode (left) or nc886-aligned counts (C) using 'FeaturePlot' and 'ViolinPlot'.

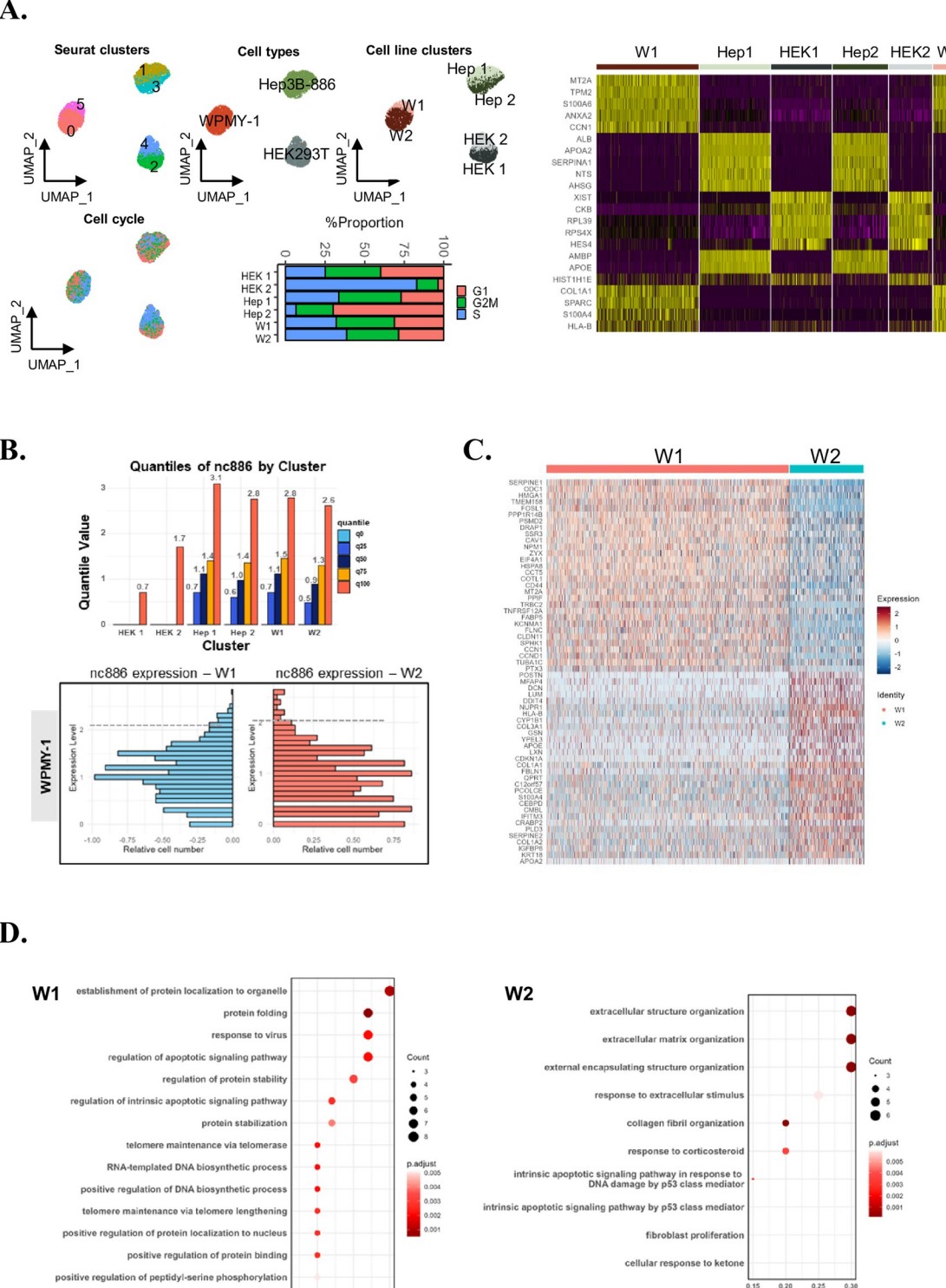

**Fig 4. Phenotypic features and nc886 expression levels in cell line sub-clusters.** (A) UMAP visualization showing two distinct clusters per cell line and their cell cycle phases. The bar plot indicates the proportion of cell cycle distribution in each of the 6 clusters. (Cluster0, W1;Cluster 1,Hep1;Cluster 2,HEK1;Cluster 3, Hep2; Cluster 4, HEK2; Cluster 5, W2). The right panel represents a heatmap visualization of cluster-specific DEGs obtained from the Wilcox test (B) Distribution of nc886 expression in the cell line clusters analyzed using Quantile Values. (C) A heatmap showing gene expression in the two clusters, W1 and W2, within the

WPMY-1 cell line. (D) GSEA performed on the WPMY-1 clusters, ordered by Gene ratio and adjusted p-value. (cutoffGeneRatio >0.15, p-value <0.005, q-value <0.05).

+ mRNA at the single-cell level. The nc886 GSP included adaptor sequences necessary for sequencing library construction and oligo-tag counting in addition to sequences complementary to nc886. Initially we planned to use the oligo tag for quantifying nc886 gene expression, but we encountered a problem of significant non-specific priming. To overcome this, we selectively extracted nc886 sequences from the oligo-tagged sequences. Demonstrating the integrated analysis of nc886 and poly(A)+ mRNA in model cell lines, we confirmed the viability of our strategy. However, further optimization of the protocol is necessary.

## Improving nc886 detection in feature barcoding: Overcoming inefficiency and non-specific binding

More than 99% of the oligo tags contained sequences unrelated to nc886 (Fig 3A, right upper), indicating widespread non-specific priming. This issue may stem from the inherent characteristics of nc886, a short RNA transcribed by Pol III, which complicates GSP design. Most Pol III genes feature 4–6 consecutive thymidylates at the 3' end. As a type 2 Pol III gene, nc886 includes two intragenic promoter elements, box A and box B, each approximately 15 nucleotides long. Consequently, more than 30% of the nc886 sequence shares potential homology to several-hundred type 2 Pol III genes. This necessitates placing the GSP outside these common motifs at the 3' end. The constraints on designing an effective primer were substantial, limiting the design of multiple primers and selection of the optimal one.

While the primer we designed was predicted to be specific *in silico*, it appeared in actual scRNA-seq experiments that this primer bound to other RNAs that share similar sequences with nc886. The loop-shaped secondary structure of nc886 may have contributed to the low specificity as well [39, 40]. Synthesizing cDNA at high temperatures using a thermophilic reverse transcriptase may reduce secondary structure problems.

Transitioning to a hybridization approach instead of primer extension could also potentially yield better results in addressing non-specificity issues. The Fixed RNA Seq method developed by 10X Genomics uses multi-probe hybridization techniques for gene expression analysis, which could be adapted for detecting nc886. However, the limited options for primer design remain a major obstacle, especially given our aim to establish a foundation for analyzing other Pol III genes and ultimately obtaining single-cell Pol III transcriptomes.

## Analyzing variability in nc886 expression and its effect on gene expression in cell lines

Previous studies have employed microwell-based methods for single-cell sequencing of total RNAs [11–13, 41]. In these studies, cell type specific expressions were demonstrated for protein-coding genes, lncRNAs, and other ncRNAs such as miRNA, scaRNA, snoRNA, tRNA, and histone RNAs, suggesting differential regulation of a broad spectrum RNAs. Although these methods are robust, they are costly and capture a relatively small number of cells.

In contrast, we chose the droplet-based oligo-tag method for its cost efficiency and scalability. This method enabled us to perform an integrated analysis of 5' scRNA-seq and nc886 in cell lines, serving as a proof of concept for other ncRNAs. However, within the same cell line population, the differences in nc886 expression were not pronounced enough to observe clear distinctions between subpopulations, which was likely due to the homogeneous nature of cell lines resulting in limited biological insights.

Optimizing the amplification process with more diverse cell lines and primary samples could be beneficial to gain clearer insights into the functions of nc886. Using a wide variety of samples could help elucidate the differential expression and regulation of nc886 across cellular contexts and conditions. Furthermore, employing additional techniques such as high-temperature cDNA synthesis with thermophilic reverse transcriptases and rRNA depletion methods could enhance the specificity and efficiency of ncRNA capture, potentially providing more meaningful biological insights.

## Conclusions

We modified the 5' scRNA-seq protocol by incorporating an oligo-tagged nc886 GSP during the cDNA synthesis, which allowed simultaneous analysis of non-polyA-tailed nc886 and gene expression data. While further protocol optimization is required, this primer spike-in strategy could be used for the detection of other genes without polyA tails and their impact on the gene expression phenotype.

## Supporting information

**S1 Fig. In vitro validation of scRNA-seq protocol for nc886 detection.** (A) Schematic diagram of recapitulation of the scRNA-seq steps in vitro. Total RNA isolated from Huh7, a hepatoma cell line expressing nc886 abundantly, was subjected to cDNA synthesis and PCR amplification, with or without the cDNA amplification step. Primer sequences (5' to 3') are: Read 1, ctacacgacgctcttccgatct; nc886 101–80, aagggtcagtaagcacccgcg; 5'-nested, ccgatctaaacctgagaaacc; nc886 89–70, ggtctcgaaccccagcacag. (B-C) 2% agarose gel electrophoresis and ethidium bromide staining visualization (panel B) and melting curves (panel C) of the nest PCR product. The 1st PCR was performed with LightCycler 480 SYBR Green I MasterMix (Roche, Penzberg, Germany), with cDNA from 20 ng of total RNA. The nested PCR was done with LightCycler 480 SYBR Green I MasterMix (Roche, Penzberg, Germany), using 1/200 (the + cDNA amplification sample) or 1/1000 (the—cDNA amplification sample) of the 1st PCR product. In panel B, the molecular sizes of 100bp Opti-DNA Marker (Applied Biological Materials; Richmond, Canada) are shown on the left. Lane 2 is blank.
(PDF)

**S1 Raw images.**
(PDF)

## Author Contributions

**Conceptualization:** Nayoung Kim, Huiram Kang, Dongwan Hong, Jiyoung Joan Jang, Hwi-Ho Lee, Hae-Ock Lee.

**Data curation:** Gyeong-Jin Shin, Byung-Han Choi, Areum Jo, Nayoung Kim.

**Formal analysis:** Hye Hyeon Eum, Areum Jo, Yeon-Su Lee.

**Investigation:** Gyeong-Jin Shin, Hye Hyeon Eum, Areum Jo, Huiram Kang, Jiyoung Joan Jang.

**Methodology:** Nayoung Kim, Dongwan Hong, Hwi-Ho Lee.

**Project administration:** Yong Sun Lee, Hae-Ock Lee.

**Supervision:** Yong Sun Lee, Hae-Ock Lee.

**Validation:** Byung-Han Choi.

**Visualization:** Gyeong-Jin Shin, Areum Jo.

**Writing – original draft:** Gyeong-Jin Shin.

**Writing – review & editing:** Byung-Han Choi, Yong Sun Lee, Hae-Ock Lee.

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
