## [Decision Letter · Decision Letter 0]

6 May 2024

PONE-D-24-10785Single cell RNA sequencing of nc886, a non-coding RNA transcribed by RNA polymerase III, with a primer spike-in strategyPLOS ONE

Dear Dr. Lee,

Thank you for submitting your manuscript to PLOS ONE. After careful consideration, we feel that it has merit but does not fully meet PLOS ONE’s publication criteria as it currently stands. Therefore, we invite you to submit a revised version of the manuscript that addresses the points raised during the review process.

We look forward to receiving your revised manuscript.

Kind regards,

Abozar Ghorbani, Ph.D

Academic Editor

PLOS ONE

Journal Requirements:

When submitting your revision, we need you to address these additional requirements. 1. Please ensure that your manuscript meets PLOS ONE's style requirements, including those for file naming. The PLOS ONE style templates can be found at https://journals.plos.org/plosone/s/file?id=wjVg/PLOSOne_formatting_sample_main_body.pdf and https://journals.plos.org/plosone/s/file?id=ba62/PLOSOne_formatting_sample_title_authors_affiliations.pdf 2. Please note that funding information should not appear in any section or other areas of your manuscript. We will only publish funding information present in the Funding Statement section of the online submission form. Please remove any funding-related text from the manuscript. 3.  Please note that your Data Availability Statement is currently missing the repository name and/or the DOI/accession number of each dataset OR a direct link to access each database. If your manuscript is accepted for publication, you will be asked to provide these details on a very short timeline. We therefore suggest that you provide this information now, though we will not hold up the peer review process if you are unable. 4. Please include your tables as part of your main manuscript and remove the individual files. Please note that supplementary tables (should remain/ be uploaded) as separate "supporting information" files. 5. PLOS ONE now requires that authors provide the original uncropped and unadjusted images underlying all blot or gel results reported in a submission’s figures or Supporting Information files. This policy and the journal’s other requirements for blot/gel reporting and figure preparation are described in detail at https://journals.plos.org/plosone/s/figures#loc-blot-and-gel-reporting-requirements and https://journals.plos.org/plosone/s/figures#loc-preparing-figures-from-image-files. When you submit your revised manuscript, please ensure that your figures adhere fully to these guidelines and provide the original underlying images for all blot or gel data reported in your submission. See the following link for instructions on providing the original image data: https://journals.plos.org/plosone/s/figures#loc-original-images-for-blots-and-gels.   In your cover letter, please note whether your blot/gel image data are in Supporting Information or posted at a public data repository, provide the repository URL if relevant, and provide specific details as to which raw blot/gel images, if any, are not available. Email us at plosone@plos.org if you have any questions. 

Reviewers' comments:

Reviewer's Responses to Questions

**Comments to the Author**

1. Is the manuscript technically sound, and do the data support the conclusions?

Reviewer #1: No

Reviewer #2: No

2. Has the statistical analysis been performed appropriately and rigorously? 

Reviewer #1: I Don't Know

Reviewer #2: No

3. Have the authors made all data underlying the findings in their manuscript fully available?

Reviewer #1: Yes

Reviewer #2: Yes

4. Is the manuscript presented in an intelligible fashion and written in standard English?

Reviewer #1: Yes

Reviewer #2: No

5. Review Comments to the Author

Reviewer #1: I have questions regarding the experimental design that require clarification from the authors?

1.Limited ncRNA Coverage: While the developed methodology represents a significant step in addressing the challenge of profiling non-coding RNAs (ncRNAs) transcribed by RNA polymerase III (Pol III), it may still have limitations in capturing all types of ncRNAs. It's possible that other classes of ncRNAs or specific transcripts may not be effectively captured using this approach. How do the authors plan to address these potential limitations?

2. Oligo Tag Specificity: The authors note that the identification of oligo tags showed limited target specificity, implying that the oligo-tagged primer approach may not be completely specific to the target ncRNA. This raises concerns about potential off-target effects or noise in the sequencing data. Have the authors encountered off-target effects during their experimentation? How did they evaluate the specificity of the tags for other ncRNAs?

3. Validation and Reproducibility: While the methodology was applied to three different cell lines with varying levels of nc886 expression, further validation across a larger and more diverse set of cell types, tissues, or biological conditions would enhance the robustness and generalizability of the findings. Additionally, ensuring reproducibility of the results across different experimental setups and laboratories is essential. The authors' validation of this approach and the reproducibility of the results are important aspects to consider. How authors will address it?

Reviewer #2: The manuscript entitled “Single cell RNA sequencing of nc886, a non-coding RNA transcribed by RNA polymerase III, with a primer spike-in strategy” have provided a measurement protocol for nc886 and then applied this protocol in three cell lines which express high, low, and zero levels of nc886.

While it is a very valuable and interesting work, the manuscript raises several serious concerns to support its conclusion:

1. The authors must enhance the manuscript language by getting it reviewed by an English language expert to make it convenient for the readers to comprehend. The manuscript needs many corrections in terms of grammar and spelling.

2. In the method, the author should specify the cell type used for each cell line and the primers utilized in the Real-time PCR.

3. The discussion is inadequately written. It is essential to compare the results with previous studies.

4. What are the planned future experiments?

5. The complete results section description needs to be detailed.

6. The protocol's validation should involve a larger number of cell lines.

7. At the end of the manuscript, the conclusion is not provided.

6. PLOS authors have the option to publish the peer review history of their article (what does this mean?). If published, this will include your full peer review and any attached files.

Reviewer #1: No

Reviewer #2: No

---

## [Author Response · Author response to Decision Letter 0]

19 Jun 2024

June 14, 2024

Abozar Ghorbani, PhD

Dear Dr. Abozar,

We would like to thank you and the reviewers for the valuable feedback on our manuscript titled “Single cell RNA sequencing of nc886, a non-coding RNA transcribed by RNA polymerase III, with a primer spike-in strategy”. We have carefully considered the comments and suggestions and revised the manuscript accordingly.

Reviewer #1:

I have questions regarding the experimental design that require clarification from the authors? 

1.Limited ncRNA Coverage: While the developed methodology represents a significant step in addressing the challenge of profiling non-coding RNAs (ncRNAs) transcribed by RNA polymerase III (Pol III), it may still have limitations in capturing all types of ncRNAs. It's possible that other classes of ncRNAs or specific transcripts may not be effectively captured using this approach. How do the authors plan to address these potential limitations?

: We agree with the reviewer that there are limitations and challenges in capturing other RNAs. Our method is not suitable for RNAs that cannot be amplified with gene specific primers. We chose the strategy of barcoded gene-specific primers (GSPs) because of its simplicity and expandability. This strategy can be applied to other ncRNAs by simply using different GSPs. Therefore, we expect that our experience with nc886 will provide valuable insights and tips for working with other ncRNAs. In our scRNA-seq of nc886, we encountered non-specific priming, but were able to map the sequencing reads successfully. Our approach to overcoming this issue can be applied if a similar problem arises in scRNA-seq of other ncRNAs. Additionally, our study clearly highlighted the need to improve the scRNA-seq protocol. In our revised manuscript, we have extensively discussed potential directions and solutions for improvement (pages 14-15, lines 287-306 in the revised version), all of which can be applied to other classes of ncRNAs. 

2. Oligo Tag Specificity: The authors note that the identification of oligo tags showed limited target specificity, implying that the oligo-tagged primer approach may not be completely specific to the target ncRNA. This raises concerns about potential off-target effects or noise in the sequencing data. Have the authors encountered off-target effects during their experimentation? How did they evaluate the specificity of the tags for other ncRNAs?

: The off-target effects were demonstrated by the sequencing data (Figure 3A, right side: upper bar graph), showing that only a small fraction of the feature barcode counts corresponded to the nc886 gene. Our in vitro experiment (newly added Figure S1) also indicated while primers bound to nc886, non-specific priming was amplified during the cDNA amplification step. We addressed this by extracting sequences located after the feature barcode and counting those mapped to nc886. As detailed in lines 287-295 on pages 14-15 of the revised version, primer design is for nc886 is particularly limited and challenging. For other ncRNAs, especially those of sufficient length, the best strategy is to design multiple primers, select the best one through preliminary in vitro experiments (such as Figure S1), and then proceed with scRNA-seq. In the worst case, as our experience with nc886 suggests, sequences can be extracted. 

3. Validation and Reproducibility: While the methodology was applied to three different cell lines with varying levels of nc886 expression, further validation across a larger and more diverse set of cell types, tissues, or biological conditions would enhance the robustness and generalizability of the findings. Additionally, ensuring reproducibility of the results across different experimental setups and laboratories is essential. The authors' validation of this approach and the reproducibility of the results are important aspects to consider. How authors will address it?

: As the reviewer pointed out, we also recognize that it is important to apply this technique to many cell types, tissues, and biological conditions to enhance the robustness and generalizability of our findings. Collecting and comparing more data will not only provide biological information, but also help validate the modified scRNA protocol. However, we are more focused on optimizing the protocol before performing scRNA-seq in many samples. Once optimized, we plan to perform scRNA-seq experiments in many biological samples.

Reviewer #2: The manuscript entitled “Single cell RNA sequencing of nc886, a non-coding RNA transcribed by RNA polymerase III, with a primer spike-in strategy” have provided a measurement protocol for nc886 and then applied this protocol in three cell lines which express high, low, and zero levels of nc886.

While it is a very valuable and interesting work, the manuscript raises several serious concerns to support its conclusion:

1. The authors must enhance the manuscript language by getting it reviewed by an English language expert to make it convenient for the readers to comprehend. The manuscript needs many corrections in terms of grammar and spelling.

: In accordance with the reviewers' comments, our manuscript has been edited by a professional English editing service. The certificate is attached.

2. In the method, the author should specify the cell type used for each cell line and the primers utilized in the Real-time PCR.

: We specified the cell type used for each cell line and the primers utilized in the Real-Time (RT) PCR (in the revised Fig. 1A). 

3. The discussion is inadequately written. It is essential to compare the results with previous studies.

: We have reorganized and rewritten the discussion into three sections. In the first section, we have summarized our findings, highlighting both the benefits and the limitations. In the second section, we have discussed the limitations in detail and proposed potential strategies to address them. The final section compares our study with previous single-cell studies on ncRNAs and discusses further research directions and potential extensions. 

4. What are the planned future experiments?

: Based on the points discussed in the Discussion section, we are planning experiments to address the current limitations identified. 

One significant challenge is the inefficient amplification of nc886-GSP products. We have proposed several strategies for improvement to mitigate issues of non-specificity (lines 296-306, page 15). These strategies are designed to enhance the efficiency of capturing ncRNA.

After optimizing the protocol, we intend to apply our method to a broad spectrum of cell types, tissues, and biological conditions. Specifically, we are interested in exploring the role of nc886 in colon cancer and investigating how its heterogeneous expression correlates with the gene expression phenotypes of tumor cells.

5. The complete results section description needs to be detailed.

: In response to the reviewer’s critique, we revised the Results section to provide clearer descriptions of the figures. Specific changes include: 

Page 10, lines 180-181

The cell lines were specifically mentioned (WPMY-1, Hep3B-nc886, HEK293T), and detailed descriptions of the cell lines were added.

Page 10, lines 182-184 

Methods for verifying nc886 expression levels using qRT-PCR and Northern hybridization were added.

Page 10, lines 185-195

We provided comprehensive information about nc886-GSP and detailed the in vitro validation performed before implementing the modified scRNA-seq.

Page 10, lines 194-196

We included information confirming the correct synthesis of the nc886 cDNA, as verified by PCR results and sequencing, which are shown in the revised supplementary Fig. S1. These revisions offer a clearer and more specific explanation to assess the validity of nc886 detection and support the feasibility of using GSPs.

Page 12, lines 231-247 

We updated the Fig 3A and improved the alignment between the figure and the text descriptions in detail.

Page 13, lines 249-250 (title), 251-271 (contents) 

We modified the section title and emphasized the integrated analysis of nc886 and 5’ gene expression data. 

page 13, lines 268-271 

We proposed that the heterogeneity of nc886 expression levels might also be relevant to other samples, such as colon cancer, offering additional biological insights.

For additional changes, please refer to the tracked version of the revised manuscript. 

6. The protocol's validation should involve a larger number of cell lines.

We agree with the reviewer that this protocol requires further validation with a larger number of cell lines. Expanding the range of cell lines will enable us to quantify the gene expression levels of nc886 and match them with single-cell analysis. We added a discussion section on this point (page 16, lines 320-322 in the revised version).

7. At the end of the manuscript, the conclusion is not provided.

In the revision, we added a Conclusion section after the Discussion. (page 16, lines 328-333 in the revised version).

---

## [Decision Letter · Decision Letter 1]

8 Jul 2024

Single cell RNA sequencing of nc886, a non-coding RNA transcribed by RNA polymerase III, with a primer spike-in strategy

PONE-D-24-10785R1

Dear Dr. Lee,

We’re pleased to inform you that your manuscript has been judged scientifically suitable for publication and will be formally accepted for publication once it meets all outstanding technical requirements.

Kind regards,

Abozar Ghorbani, Ph.D

Academic Editor

PLOS ONE

Additional Editor Comments (optional):

Reviewers' comments:

Reviewer's Responses to Questions

**Comments to the Author**

1. If the authors have adequately addressed your comments raised in a previous round of review and you feel that this manuscript is now acceptable for publication, you may indicate that here to bypass the “Comments to the Author” section, enter your conflict of interest statement in the “Confidential to Editor” section, and submit your "Accept" recommendation.

Reviewer #1: All comments have been addressed

Reviewer #2: All comments have been addressed

2. Is the manuscript technically sound, and do the data support the conclusions?

Reviewer #1: Partly

Reviewer #2: Partly

3. Has the statistical analysis been performed appropriately and rigorously? 

Reviewer #1: Yes

Reviewer #2: Yes

4. Have the authors made all data underlying the findings in their manuscript fully available?

Reviewer #1: Yes

Reviewer #2: Yes

5. Is the manuscript presented in an intelligible fashion and written in standard English?

Reviewer #1: Yes

Reviewer #2: Yes

6. Review Comments to the Author

Reviewer #1: The author's responses were highly compelling, and this data is sufficient for publication in support of the newly developed protocol. This manuscript is well-written and does not require further revision.

Reviewer #2: (No Response)

7. PLOS authors have the option to publish the peer review history of their article (what does this mean?). If published, this will include your full peer review and any attached files.

Reviewer #1: No

Reviewer #2: No

---

## [Editor Report · Acceptance letter]

16 Aug 2024

PONE-D-24-10785R1 

PLOS ONE

Dear Dr. Lee, 

I'm pleased to inform you that your manuscript has been deemed suitable for publication in PLOS ONE. Congratulations! Your manuscript is now being handed over to our production team.

Kind regards, 

on behalf of

Dr. Abozar Ghorbani 

Academic Editor

PLOS ONE